# Application of Ceramic Lattice Structures to Design Compact, High Temperature Heat Exchangers: Material and Architecture Selection

**DOI:** 10.3390/ma14123225

**Published:** 2021-06-11

**Authors:** Marco Pelanconi, Simone Zavattoni, Luca Cornolti, Riccardo Puragliesi, Edoardo Arrivabeni, Luca Ferrari, Sandro Gianella, Maurizio Barbato, Alberto Ortona

**Affiliations:** 1Mechanical Engineering and Materials Technology Institute (MEMTi), University of Applied Sciences (SUPSI-DTI), Polo Universitario Lugano, 6962 Lugano, Switzerland; simone.zavattoni@supsi.ch (S.Z.); luca.cornolti@supsi.ch (L.C.); riccardo.puragliesi@supsi.ch (R.P.); edoardo.arrivabeni@supsi.ch (E.A.); maurizio.barbato@supsi.ch (M.B.); alberto.ortona@supsi.ch (A.O.); 2Department of Industrial Engineering, University of Padova, 35131 Padova, Italy; 3EngiCer SA, 6828 Balerna, Switzerland; luca@engicer.com (L.F.); sandro@engicer.com (S.G.)

**Keywords:** heat exchanger, material selection, lattice structure, silicon carbide, CFD

## Abstract

In this work, we report the design of ceramic lattices produced via additive manufacturing (AM) used to improve the overall performances of compact, high temperature heat exchangers (HXs). The lattice architecture was designed using a Kelvin cell, which provided the best compromise among effective thermal conductivity, specific surface area, dispersion coefficient and pressure loss, compared to other cell geometries. A material selection was performed considering the specific composition of the fluids and the operating temperatures of the HX, and Silicon Carbide (SiC) was identified as promising materials for the application. The 3D printing of a polymeric template combined with the replica method was chosen as the best manufacturing approach to produce SiC lattices. The heat transfer behaviour of various lattice configurations, based on the Kelvin cell, was determined through computational fluid dynamics (CFD). The results are used to discuss the application of such structures to compact high temperature HXs.

## 1. Introduction

Equipment that works at high temperature (above 1000 °C) must be designed by investigating materials, structure and manufacturing processes that can ensure properties and performance required by the application, such as thermal stability, thermal conductivity, thermal expansion, service temperature resistance, oxidation resistance, chemical stability, etc. Such components can be found in different industrial plants in the form of heat exchangers, reactors, burners, solar receivers, heat storage systems and so on. Thanks to the rapid development of additive manufacturing (AM) technologies, both software and hardware, it is now possible to enhance the efficiency of these components by designing compact new generation structures.

Several approaches have been investigated to improve the performance of high temperature heat exchangers (HXs) [1,2] and, in recent years, complex porous architectures (such as lattices) have received significant interest due to their tunable multifunctional properties [3,4,5,6]. A lattice consists of a periodic arrangement of a unit-cell, made up of cylindrical struts connected to each other [7]. The morphology of the unit-cell can be designed and varied according to the final function of the component and its performances greatly depends on it [8,9]. The design method is usually based on purpose-built algorithms or software that can generate lattice structures with several parametric variables, such as cell type, cell size, cell distortion, struts diameter and struts distortion [10,11,12]. Several types of unit-cells (cube, rotated cube, kelvin, octet, crystal, star, etc.) have been investigated finding that each offers very different properties from the others, such as specific surface area, porosity, pressure drop, effective thermal conductivity, heat transfer coefficient, dispersion coefficient and fluid mixing [13,14,15,16,17].

However, the design of the lattices also depends on the manufacturing technique used. There are different AM approaches to fabricate lattices and the choice between them depends mainly on the material properties. In high temperature applications, ceramic lattices are widely used as they withstand both high temperature and oxidation. The choice of the most suitable ceramic material can be performed using different approaches and the most popular is the Ashby’s material selection [18], which consists in the comparison between the materials properties required by the application. To the best of the authors’ knowledge, three main ceramics are commonly used to fabricate lattices for high temperature applications [19,20,21]: Alumina (Al_2_O_3_), Zirconia (ZrO_2_) and Silicon Carbide (SiC).

The fabrication of ceramics components can be performed through several different AM technologies, which can be divided into direct and indirect methods. Direct AM consists in the fabrication of the part using ceramic raw materials and obtaining directly ceramic or green object. Examples of direct AM technology are: stereolithography, selective laser sintering, binder jetting, fused deposition modeling, laminated object manufacturing, robocasting and direct ink writing [22,23,24]. These approaches are commonly followed by cleaning, de-binding, sintering followed by post-processing. One hybrid AM approach consists in the fabrication of a polymeric template through common additive manufacturing technologies, followed by the processing of the ceramic or green part with traditional methods, such as slurry infiltration, replica method, polymer precursor infiltration, slip or gel casting, chemical or physical vapor deposition, liquid silicon infiltration and so on [19,25,26]. Each technique allows to obtain a material with different resolution (coarse or fine details) and different material properties. Therefore, the definition of the lattice morphology, the material selection and the choice of manufacturing technique have become crucial aspects for the design of new generation HXs.

In this work, we aim to introduce the development of an innovative high temperature heat exchanger to be coupled with the solar process of the HYDROSOL-beyond project (http://www.hydrosol-beyond.certh.gr, accessed on 30 April 2021). The purpose of this project is the utilisation of concentrated solar thermal power for the production of Hydrogen from the dissociation of water via the redox-pair-based thermochemical cycles. HYDROSOL-beyond is the continuation of a previous Fuel Cells and Hydrogen Joint Undertaking (FCH-JU) project, the HYDROSOL-Plant, which implemented and tested the operation of a 750 kWth solar plant. Further aim of the HYDROSOL-beyond project is to boost the performance of the current technology through innovative solutions that will increase the potential of the technology’s future commercialisation.

Compact gas-to-gas HXs have been designed for maximising the heat recovery from the high-temperature gaseous stream, leaving the receiver, with the aim of preheating the incoming gaseous flow into the receiver. Major novelty has been the utilisation of engineered ceramic cellular architectures for enhancing the efficiency of the heat transfer thanks to the favourable convection and radiation phenomena induced by such structures. They have been arranged between two consecutive plates which constitute the border of a channel in which the fluid flows. Figure 1 shows the schematic of the proposed compact gas-to-gas heat exchangers. The design of the lattice included the material selection and morphology, and numerical simulations were used to investigate its behaviour in heat transfer.

## 2. Materials and Methods

### 2.1. HXs Operating Conditions

Table 1 shows the operating conditions of the two high temperature heat exchangers to be developed for the purposes of the HYDROSOL-beyond project.

### 2.2. Ashby-Based Approach

The Ashby approach [18] was used to help in the materials selection of the two high-temperature HXs based on their boundary and operating conditions. This method consisted in the identification of the system’s constraints and objectives that the materials must meet. The constraints are the operating conditions of the HXs (such as fluid composition and fluid temperature) and they have to be imposed to the materials database in order to ignore the materials that cannot meet the requirements. The objectives are the material properties that can be plotted in diagrams, isolating the subset of materials that are suited for the design. In the material choice (Ansys GRANTA EduPack software, ANSYS, Inc., Cambridge, UK, 2021 (www.ansys.com/materials, accessed on 30 April 2021)), the following properties were evaluated: maximum service temperature, thermal conductivity, thermal shock resistance, thermal expansion, heat capacity, strength, modulus, manufacturability and cost. The materials properties were plotted one as a function of another, mapping out the fields in property space occupied by each material class and the subfields occupied by individual materials. The resulting charts were useful because they condensed a large amount of information into a compact form and revealed correlations between the materials.

### 2.3. Design

In compact gas-to-gas heat exchangers, the channels in which hot and cold gas alternatively flows are piled up to form the core of the heat exchanger. In one of the most common configurations, these channels have a rectangular cross section. In this application, the twofold aim of the lattices is to enhance the flow mixing and to increase the heat transfer area. Therefore, they have to be arranged between two consecutive plates which constitute the border of a channel in which the fluid flows (Figure 1). As previously anticipated, their presence disrupts the trajectory of the fluid, promoting the generation of a thin boundary layer close to the solid surface and therefore, they enhance heat transfer because of higher convective coefficients.

The lattice architecture was designed with a purpose-built algorithm using Matlab R2019a (MathWorks, Natick, MA, USA). The 3D numerical tool contains a library of unit-cells, which can be selected and replicated in the space, in order to form the lattice structure. Several parameters can be set: cell type, cell size, cell size gradient, cell type gradient, struts diameter, struts gradient, distortion coefficient, lattice shape and dimensions [10,11]. The choice of the cell type and size was based on the literature review and manufacturing constraints.

### 2.4. Additive Manufacturing

For the manufacturing of ceramic periodic structures, the lattice geometry was designed, dimensioned and printed through additive manufacturing in polymeric material. The printed polymeric structures served as templates for the ceramic coating process. The templates were impregnated with a SiC-based ceramic slurry, excess slurry was removed and the operation was repeated until sufficient thickness is built. The coated template structure (green body) was then dried, pyrolyzed in inert atmosphere up to 1000 °C and liquid silicon infiltrated at 1500 °C, delivering the final SiSiC periodic cellular structure. Figure 2 shows the main production steps employed. Table 2 shows the pros and cons of the replica method used to fabricate the SiSiC lattices.

### 2.5. Numerical Simulations

#### 2.5.1. Heat Transfer Characterisation

This part of the work aims to provide useful data for the design of advanced heat exchangers which exploits lattice structures. In particular, the objective is to define some correlations, derived from the results of accurate computational fluid dynamics (CFD) simulations, which can be applied to different design tools such as: (i) the ε-NTU 0D approach [27], (ii) 1D heat transfer codes [28] or (iii) multiscale 3D computation of heat exchangers in which the core is modelled as a porous media and the geometry of the lattice is not directly resolved [29]. To this end, a proper definition of quantities and simulation setups can help to decouple the effect of the many parameters which influence heat transfer (flow properties, geometry, material properties, temperature, etc.) and contributes to generalise the results. Thus, in this section, the definition of the quantities extrapolated from CFD simulations results and some modelling considerations are provided.

When the fluid absorbance is negligible, the main parameter which characterises the heat transfer between a fluid flow and a solid geometry is the convective heat transfer coefficient h [27]. To decouple the effect of the lattice material properties from the one related to the interaction between fluid flow and the selected geometry, in the first simulations campaign only the fluid region was solved, while the lattice surface was modelled as a boundary conditions with uniform temperature. This implies that the lattice behaves like an ideal fin and the thermal power removed, in this case, is labelled ideal (${\dot{Q}}_{\mathrm{id}}$). For the investigated setup, the appropriate formula to estimate the average convective heat transfer coefficient is [30]:(1)$$h=\frac{{\dot{Q}}_{\mathrm{id}}}{\Delta T_{\mathrm{ml}}\cdot A_{\mathrm{HE},\mathrm{tot}}}$$
where ${\dot{Q}}_{\mathrm{id}}$ is the heat rate removed from the fluid, $A_{\mathrm{HE},\mathrm{tot}}$ is the total heat transfer area and ΔT_ml_ is the logarithmic mean temperature difference [30].

It is well known that the influence of flow and fluid properties on convective heat transfer can be combined into two non-dimensional groups which are the Reynolds Re and Prandtl Pr numbers, respectively [27], while the convective heat transfer coefficient is recast in terms of Nusselt number Nu:(2)$$\mathrm{Nu}=\frac{{h\;L}_{\mathrm{ref}}}{k_{f}}$$
where k_f_ is the fluid thermal conductivity. The selected reference length L_ref_ is the Kelvin cell edge (Figure 5a), which is the equivalent of the average porous diameter usually employed in the literature [31]. This length is used also to define the Reynolds number.

Consequently, in the first simulations campaign, the Reynolds and Prandtl numbers were varied within the following ranges: 5 < Re < 215 and 0.7 < Pr < 0.9. These ranges represent the actual operative working points of the compact gas-to-gas heat exchanger at high temperature under investigation. In particular, this range of Reynolds numbers approximately corresponds to velocities between 0.1 m/s and 5.0 m/s when the fluid is Nitrogen at high temperature levels (about 1200 °C).

The effect of the lattice material thermal conductivity on the heat transfer was investigated with a second simulations campaign where both the fluid and the solid domains, the latter corresponding to the volume occupied by the lattice, were considered. In this campaign, the radiative heat transfer was not taken into account; its effect is investigated in the third simulations campaign. The influence of the material properties is embedded within the global lattice efficiency which is defined as:(3)$$\eta_{g}=\frac{\dot{Q}}{{\dot{Q}}_{\mathrm{id}}}$$
where $\dot{Q}$ is the resulting heat transfer rate computed with this simulations campaign, while ${\dot{Q}}_{\mathrm{id}}$ is the same quantity computed in the first simulations campaign. By definition, the fin efficiency can be computed as [27]:(4)$$\eta_{\mathrm{fin}}=1-\frac{A_{\mathrm{HE},\mathrm{tot}}}{A_{\mathrm{fin}}}\left(1-\eta_{g}\right)$$
where $A_{\mathrm{fin}}$ is the surface area of the lattice in contact with the fluid. This efficiency takes into account the fact that, moving away from the base, the lattice temperature tends to reach the fluid temperature, which in turn decreases the heat transfer rate with respect to the ideal case. In this campaign two parameters were varied: the average velocity of the flow and the thermal conductivity of the solid material. Indeed, once the geometry is fixed, the heat transfer theory suggests that the fin efficiency dependents on the convective heat transfer coefficient and the thermal conductivity. Specifically, three values of thermal conductivity were investigated: 2 W/(m∙K), 6 W/(m∙K) and 46 W/(m∙K). The first value is representative of ZrO_2_ [32], the second value is representative of Al_2_O_3_ [32] while the last value is representative of Si-SiC [33] at high temperatures. The sub-set of the flow properties of the first simulations campaign, for which the Prandtl number is equal to 0.7, was employed in this second one.

Heat transfer theory suggests combining the convection coefficient h, the thermal conductivity of the solid k and the geometry of the fin into the following non-dimensional group [27]:(5)$$\mathrm{mL}=\sqrt{\frac{h\;p}{k\;A}}\cdot L_{\mathrm{fin}}=\sqrt{\frac{4\;h}{{k\;d}_{\mathrm{strut}}}}L_{\mathrm{fin}}$$
where p/A represents the ratio of the fin cross section perimeter over the cross-section area; $L_{\mathrm{fin}}$ represents the fin length and $d_{\mathrm{strut}}$ is the strut diameter. For the lattice structure under investigation, the reference geometry used to compute the p/A ratio is the cylinder, replicating the struts topology, while the length of the fin was estimated as $L_{\mathrm{fin}}=\sqrt{2}n\cdot L_{\mathrm{ref}}/2$, where n is the number of cells along the channel height.

Finally, in the third simulations campaign, the effect of thermal radiation on the overall heat transfer was investigated. Indeed, at high temperature levels, radiation potentially plays a significant role in the heat transfer. Assuming that the gas is a non-participating (transparent) medium, such as air, and that the lattice behaves like an opaque grey body [34], the main expected effect of radiative heat transfer is to promote temperature homogeneity of the lattice, as radiative heat transfer is exchanged only among the solid surfaces with different temperatures. In particular, it makes the temperature of the lattice in the centre of the channel closer to the one of the lattice base (channel walls) which is fixed in the selected setup. Thus, the effect of radiative heat transfer is to increase the fin efficiency instead of directly influence the value of the convective heat transfer coefficient: for instance, if the solid has a uniform temperature distribution (fin efficiency equal to one), the radiative heat transfer among the solid surfaces would be identically zero and the thermal power removed by the fluid would be identical to the one investigated in the first campaign. Therefore, a lattice global efficiency can be defined as previously done in Equation (3):(6)$$\eta_{g,\mathrm{rad}}=\frac{{\dot{Q}}_{\mathrm{rad}}}{{\dot{Q}}_{\mathrm{id}}}$$
where ${\dot{Q}}_{\mathrm{rad}}$ is the heat transfer rate computed in the third simulations campaign and ${\dot{Q}}_{\mathrm{id}}$ is the heat transfer rate computed with the first simulations campaign. From this definition and the previous assumptions, this global efficiency is expected to vary within the range $\eta_{g}\leq \eta_{g,\mathrm{rad}}\leq 1$. The results of the third simulation campaign are expressed as a function of the following non-dimensional parameter:(7)$$\frac{\sigma \cdot 4T_{m}^{3}}{h}$$
where σ is the Stefan-Boltzmann constant and the linear coefficient of the radiative heat transfer is:(8)$$4T_{m}^{3}=T_{\mathrm{fin}}^{3}+T_{\mathrm{fin}}^{2}T_{\mathrm{wall}}+T_{\mathrm{fin}}T_{\mathrm{wall}}^{2}+T_{\mathrm{wall}}^{3}$$

The fin temperature (T_fin_)is estimated with the following relation:(9)$$T_{\mathrm{fin}}=T_{\mathrm{fluid}}-\eta_{\mathrm{fin}}\cdot \left(T_{\mathrm{fluid}}-T_{\mathrm{wall}}\right)$$

Equation (9) provides a rough estimation of the average lattice temperature which is used as a reference to characterise the radiative heat exchange between the lattice and the channel walls whose temperature is fixed at T_wall_.

#### 2.5.2. Friction Factor

To fully characterise the lattice structures, besides the heat transfer behaviour, the induced pressure drops have to be provided. To this end, the non-dimensional friction factor f is introduced [30]:(10)$$f=\frac{2\left(\Delta p_{\mathrm{tot}}-\Delta p_{\mathrm{channel}}\right)}{\rho_{\mathrm{in}}U_{\mathrm{in}}^{2}}\cdot \frac{\rho_{m}}{\rho_{\mathrm{in}}}\cdot \frac{A_{c}}{A_{\mathrm{HE},\mathrm{tot}}}$$

In the previous equation, $\Delta p_{\mathrm{tot}}$ is the total pressure difference, between fluid inlet and outlet sections, which is computed through the simulations. $A_{c}$ is the total transversal cross section area that does not account for the obstruction related to the lattice structure. $U_{\mathrm{in}}$ is the inlet average velocity, $\rho_{\mathrm{in}}$ is the inlet density and $\rho_{m}$ is an average density related to the exponential variation of temperature. $\Delta p_{\mathrm{channel}}$ is the pressure loss related to the artificial extension of the computational domain (see Figure 3a), from the inlet and outlet regions, meant to provide enough room to the flow to develop the proper hydrodynamic profile. In the selected range of Reynolds numbers, the resulting flow regime is laminar, and the following relation is valid:(11)$$\Delta p_{\mathrm{channel}}=\mu \frac{12\cdot U}{H^{2}}\Delta L_{\mathrm{dev}}$$
where H is the channel height and ΔL_dev_ is the length of the development region. It was found that this pressure loss is a small fraction of the total one (about 1.5%). In general, once the geometry is fixed, the friction coefficient is only a function of the Reynolds number, which is the same used in the Nusselt correlations.

#### 2.5.3. Simulation Domain and Boundary Conditions

The three investigated lattice structures are composed by a number of cells along the three directions (height, width and length), respectively, equal to 1 × n_2_ × 3, 2 × n_2_ × 3, 3 × n_2_ × 3, where n_2_ depends on the width of the HX core. In compact HXs, the number of cells along the channel width is much larger than that along its height; thus, to reduce the computational domain, it is assumed that the flux is periodic along spanwise direction and hence three cells only are considered (n_2_ = 3). This is a reasonable assumption which well describes the behaviour of the Kelvin cells that are not immediately in contact with the lateral wall of the HX which are the majority.

In the real application, a series of lattice structures are lined up along the core axis with just a small gap among them. Therefore, the behaviour of m-consecutive lattices is not completely represented by the sum of m-isolated lattices. To assess the error introduced by this assumption, a computation in which two lattices were put in series was run and it was found out that the difference in the heat transfer coefficient was below 2%. Therefore, this simplification is acceptable. In summary, the simulated lattice geometries are, respectively: (i) 1 × 3 × 3, (ii) 2 × 3 × 3 and (iii) 3 × 3 × 3 cells. In addition to the Kelvin cells, the structures also include two small solid supports at the top and bottom of 0.15 mm thickness. Table 3 provides the main geometrical information related to these structures.

The computational domain is a parallelepiped composed of a central box, in which the lattice is located and two additional regions, before and after the box, which are added to allow the flow to fully develop before and after the lattice. Numerical tests performed on this purpose, showed that a length equal to 1.5 the Kelvin cell edge is enough to prevent the results to be affected by the inlet and outlet surfaces. The lateral walls of the box were modelled as periodic boundary conditions. The top and bottom surfaces, representing the core channels borders, were modelled as no-slip walls. Specifically, the portion of these patches associated with the two flow development regions were set adiabatic, while the temperature portion related to the central box was fixed to a uniform value T_wall_. This last condition is justified by the fact that, as an HX core is a stack of many channels, excluding the first and last, each of them experiences almost the same conditions at the top and the bottom plates. Figure 3 shows the computational domain for the case of lattice structure 2 × 3 × 3.

In the first simulations campaign, carried out to estimate the convective heat transfer coefficient, the lattice surface was treated as a no-slip wall boundary condition with the fixed temperature T_wall_. In these simulations, only the fluid domain was solved as opposed to the other simulations campaigns wherein the lattice surface becomes the interface between the fluid and the solid domains. In this last case, the surface temperature evolves accordingly with the solution in the two domains to comply the continuity of heat flux across it.

Considering the channel without the lattice, the maximum explored Reynolds number was 650 which is well below the critical value for parallel plates (about 5750 [35]) for which transition to turbulence occurs. Therefore, a parabolic velocity profile was imposed at the inlet which corresponds to a fully developed profile for the laminar regime in parallel plate channels. On the contrary, the inlet temperature was set uniform. The static pressure at the inlet is fixed to a reference value. At the outlet, a zero gradient boundary condition is applied to every quantity. On the no-slip walls, the pressure is computed from the velocity field to enforce impermeability. Fixed value of viscosity, specific heat and thermal conductivity were used for the fluid and the solid materials. The ideal gas law was used to compute the fluid density, which is the only property depending on temperature. The selected boundary conditions are summarised in Table 4.

#### 2.5.4. Simulation Parameters, Solution Algorithm and Numerical Setup

In the simulations of the first and second campaigns, the inlet pressure was set to 1.5 bar to represents non-pressurised gas-to-gas heat exchangers in which the fluid is compressed just to overcome the pressure loss of the circuit. Since high temperature applications are the target of this work, the fluid inlet temperature was set to 1200 °C. The dynamic viscosity was fixed to 5.1 × 10^−5^ Pa∙s and the specific heat was fixed to 1238 J/(kg∙K). These values are the estimated properties of N_2_ at 1200 °C. To perform the parametric analysis, the values of the average fluid inlet velocity and thermal conductivity were modified to control the Reynolds and Prandtl numbers. Since the results are expressed through non-dimensional groups, as long as radiative heat transfer is not considered, the fluid inlet temperature and the reference thermal properties are not important parameters. The temperature of the wall was set 33 °C lower than the inlet temperature base on an estimation of temperature gradient of compact heat exchangers.

For the models including radiative heat transfer, the fluid inlet temperature was changed within the range 600–1200 °C to assess its effect, while its difference with the wall temperature was kept at 33 °C. To keep the same mass flow rate, the inlet pressure was changed accordingly to maintain the same density in all the setups.

The maximum Reynolds number investigated in this work is about 215, in the literature the critical Reynold number for which the flow is considered turbulent ranges between 150–400 [31]. Therefore, the setup characterised by the maximum velocity was tested with a low Reynolds number RANS turbulence model (k-omega SST [36]). It was found that the increase of convective heat transfer coefficient with respect to the laminar case is only about 0.03%. Therefore, it was assumed that the flow is laminar, and no turbulence model was employed in the following computations.

With the selected boundary conditions and models, the problem is stationary. For the computations without radiative heat transfer, the opensource code OpenFOAM v-2006 [37] (https://www.openfoam.com, accessed on 30 April 2021) was employed. Instead, for the third simulations campaign, the commercial Fluent code v-20 from ANSYS (Ansys^®^ Fluent Academic Research, Release 20.1) was used. Results of the two software with the same setup were run and compared showing a good agreement between each other. For the first simulations campaign, in which the fluid domain only was considered, the pressure-based compressible solver rhoSimpleFoam solver was run [38]. In the second simulations campaign, where both solid and fluid domains were considered, the conjugate heat transfer solver chtMultiRegionSimpleFoam solver was selected [38]. In these computations, the LUST interpolation scheme, which is a weighted sum of a semi-implicit linear upwind and the implicit central difference schemes, was used for the convective terms and the gauss central difference scheme [39] was used for the diffusion terms. They are both second order schemes in space.

For the simulations where radiative heat transfer was considered, the surface to surface (S2S) model was exploited [40]. For all the non-adiabatic walls, the same emissivity value was set (0.7 for ZrO_2_ and 0.6 for Si-SiC). Instead, for the inlet and outlet an emissivity value of 0 was set. View factors were calculated using a face-to-face approach, without clustering. All transport equations were approximated with an upwind second-order numerical scheme.

Statistical data analysis and regressions were performed with Minitab software (Minitab: Data Analysis, Statistical and Process Improvement Tools: https://www.minitab.com, accessed on 30 April 2021).

## 3. Results and Discussion

### 3.1. Material Selection

This paragraph shows the materials selection results for the inert gas circuit heat exchanger and the steam generation circuit heat exchanger. Figure 4 shows the Ashby materials property charts for the selected material properties at 25 °C. The black dashed-lines and arrows indicate the direction that improves the performance of the materials based on the application requirements. Technical ceramics and stainless steels (red regions) were plotted because they are the only materials which can withstand the operating conditions of the two HXs. Three main ceramic materials are widely used for high temperature applications where cellular ceramics architectures (foams and lattices produced with AM) play the main role: Zirconium Dioxide (or Zirconia: ZrO_2_—green areas), Aluminum Oxide (or Alumina: Al_2_O_3_—yellow areas) and Silicon Carbide (SiC—blue areas). Many other ceramics were not plotted because they did not meet the required properties constraints, especially oxidation resistance and manufacturability through AM.

Figure 4A shows the thermal conductivity against the maximum service temperature. Due to the very high temperature of the fluid (1200 °C), technical ceramics are the only suitable materials for the inert gas circuit HX. Instead, also stainless steels can be used for the steam generation circuit HX due to the lower operating temperature (900 °C). Ceramics differ for the thermal conductivity values: Silicon carbide has higher thermal conductivity (100 W/(m °C)) that is commonly preferred in heat transfer applications, Zirconia has lower values (2 W/(m °C)) and Alumina is in the middle (30 W/(m °C)). Stainless steels can withstand lower service temperature and they have lower thermal conductivity than SiC and Al_2_O_3_. However, SiC ceramics can suffer passive oxidation due to the composition of the gaseous mixture (N_2_ + O_2_ and H_2_O + H_2_ + N_2_ for the inert gas and steam generation HXs, respectively) and to the operating conditions. Equations (12)–(14) show the reactions of SiC with oxygen, nitrogen and water vapor [41,42,43].
(12)$$\mathrm{SiC}\;\left(s\right)+{\;O}_{2}\;\left(g\right)\to {\;\mathrm{SiO}}_{2}\;\left(s\right)+\mathrm{CO}\;\left(g\right)$$
(13)$$3\mathrm{SiC}\;\left(s\right)+2N_{2}\;\left(g\right)\to {\;\mathrm{Si}}_{3}N_{4}\;\left(s\right)+3C\;\left(s\right)$$
(14)$$\mathrm{SiC}\;\left(s\right)+2H_{2}O\;\left(g\right)\to {\;\mathrm{SiO}}_{2}\;\left(s\right)+{\mathrm{CH}}_{4}\;\left(g\right)$$

At 1200 °C and oxygen pressure of 10^−3^–10^−4^ bar (which corresponds to the mixture pressure of 1–2 bar), the oxidation of SiC is passive (Equation (12)), leading to the formation of a silica (SiO_2_) layer on the SiC surface with consequent weight increase. With nitrogen (Equation (13)), silicon reacts forming a silicon nitride (Si_3_N_4_) layer on the SiC surface at temperature of 1200–1500 °C. The reaction between silicon and nitrogen in highly exothermic and the nitridation process could be self-accelerating if uncontrolled, therefore it is possible to firstly stabilise the SiC with Ni, in order to control the process and to produce silicon nitride. Silicon carbide reacts with H_2_O to yield amorphous SiO_2_ and CH_4_ above 500 °C (Equation (14)). The oxidation in water vapor is more effective with respect to the one in air. The main advantage of Zirconia and Alumina is that they do not oxidise. Figure 4B shows the density against the specific heat capacity. The purpose of the diagram is to identify the materials with lower specific heat capacity and lower density or the material that can be heated with less energy per unit of volume (product between the two properties). The product results showed that Silicon Carbide is the material with the lower heat capacity (2100 kJ/(m^3^K)), while Zirconia and Alumina have higher values (2700 and 3000 kJ/(m^3^K), respectively). Figure 4C shows the thermal expansion coefficient against the thermal shock resistance. The purpose of the diagram is to identify the materials with higher thermal shock resistance and lower thermal expansion coefficient. Moving from the lower-right corner to the upper-left corner, it is possible to identify the materials in order of increasing performances: from the materials with low thermal shock resistance and high thermal expansion (not good) to the materials with high thermal shock resistance and low thermal expansion (good). Results shows that: (i) Alumina has relatively high expansion coefficient and low thermal shock resistance (4.5–11 µstrain/°C and 50–230 °C, respectively); (ii) the majority of Zirconia ceramics have high expansion coefficient and medium-high thermal shock resistance (5–10 µstrain/°C and 100–340 °C, respectively), while Zirconia(H) has very low expansion coefficient (2.5 µstrain/°C) and very high thermal shock resistance (500 °C); (iii) Silicon carbide shows the higher performance having low thermal expansion (3–4 µstrain/°C) and high thermal shock resistance (120–310 °C). Stainless steels have higher thermal expansion coefficient with respect to ceramics. Figure 4D shows the Young’s modulus against the yield strength (elastic limit). The purpose of the diagram is to identify the materials with higher Young’s modulus and higher yield strength. In general, SiC ceramics have higher mechanical performances with respect to other ceramics and stainless steels (except for N-SiC).

Among the investigated material, the SiC family was identified as the best material family for the application. In general, SiC ceramics have very good thermal and mechanical properties with respect to Zirconia and Alumina. Therefore, the final choice of the best material must be done between several SiC types. As previously mentioned, the maps represent only the materials that can be fabricated with AM and reaction bonded silicon carbide (Si-SiC) was chosen among them. It has the higher thermal conductivity and thermal shock resistance, lower thermal expansion and heat capacity, which are preferable properties for heat transfer applications. The big advantage of Si-SiC ceramic is that it can be produced with several AM methods (see below) and this is not possible with many other ceramics. Furthermore, the manufacturing of Si-SiC components is simpler for example than that of pure-sintered SiC. The oxidation behaviour of this ceramics should be evaluated in the author’s’ future work. The Ashby approach was useful to perform a first screening among the ceramic materials and to identify the theoretically best material solution for this application. However, the real behaviour of the materials should be experimentally tested in order to perform a proper component design.

### 3.2. Lattice Architecture

Kelvin cells (Figure 5a) were used to generate the lattices under investigation. This cell provides the best compromise between effective thermal conductivity, heat transfer coefficient, specific surface area, dispersion coefficient and pressure loss, compared to other cell geometries [44,45,46,47,48]. The Kelvin cell consists of 14 faces (6 squares and 8 hexagons), 24 vertices and 36 equal length struts. Cells were designed using a struts diameter equal to 1.10 mm and a struts length of 2.25 mm, which corresponds to a cell size of 6.35 mm (=L_ref_). These values were chosen as they represent the smallest dimensions which can be manufactured without major defects with the employed manufacturing technique and, at the same time, because small dimensions allow to achieve high specific surfaces (see below).

Three lattice configurations were investigated which distinguish each other by the number of layers of cells arranged along the height of the channel. It was assumed that the lattice fills the channel of the core along all its width. For compact heat exchanger, the channel width is much longer than its height, thus a great number of cells were expected to be placed in this direction. Along the flow direction, three consecutive cells, corresponding to a total length of 19.05 mm, were considered. This choice was made for both practical and efficiency reasons. In fact, it was hard to achieve good planar tolerances in all directions, especially the longest one that corresponds to the flow direction. Moreover, the presence of gaps along the flow direction prevented the generation of axial heat transfer which is significant for high-efficiency compact HX with solid structures made with high conductivity materials such as Si-SiC. The idea was therefore to fill the channel of a heat exchanger with a series of n-lattices along the flow direction with a gap of 2–3 millimetres among them. Figure 5 provides a view of the configurations studied in this work.

### 3.3. Replica Method

Figure 6 shows the SiSiC lattice fabricated with the replica method.

### 3.4. Heat Transfer Characterisation of the Lattice Architecture

Figure 7 shows selected results obtained for the investigated geometries. As expected, the trend of the Nusselt number with respect to the Reynolds and Prandtl is the same for the three geometries. In particular, there seems to be a transition regime for Reynolds number around 40 after which the dependence of the Nusselt on the Reynolds is almost linear, while for lower values it drops towards zero. These straight lines at upper range of investigated Reynolds numbers are shifted by a constant, while their slope is similar for the three geometries. The setups with two and three layers of cells have a closer Nusselt number compared to the other configuration, as the influence of the flat channel surface is less important with respect to the case with one lattice layer. Presumably, the results for configurations with a higher number of layers would not be very different from the ones of the three layers configuration investigated in this work.

The regression curve which was found to be the best compromise between simplicity and matching has the following form:(15)Nu=α·tanh(β·Re·Pr)γ+δ·Re·Prλ

Equation (15) is a blended strain line which has the properties to give zero Nusselt number when the Reynolds number is zero. Table 5 provides the coefficients for the three investigated geometries. The maximum error for all cases is related to the lowest investigated Reynolds and Prandtl numbers which is the point for which also the CFD results are more affected by the numerical setup. The second largest error drops below 3.7%.

Figure 8 shows the dependence of friction factor on the Reynolds number and the ratio of the Colburn Factor, which is an alternative way to express the Nusselt number, to the friction factor. This ratio is usually employed in the compact heat exchanger field to compare different geometries. It can be noticed that there is a maximum of this ratio at about Re = 30.

The Forchheimer-Darcy Equation [49] suggests the following relation between the friction factor and the Reynolds number:(16)f=ωRe+θ

Anyway, using this expression, the maximum relative error is significant for the upper range of the investigated Reynolds numbers. Hence, an alternative function which better matches the CFD results is proposed:(17)f=εRe+σ+φ·lnRe

As most codes require as input the Forchheimer-Darcy parameters, Table 6 provides the coefficient for both Equations (16) and (17). The maximum relative error drops from about 14% to about 4%.

Figure 9 shows all the CFD results related to the lattice efficiency defined in Equation (4) for the three investigated configurations and materials as a function of the non-dimensional group mL defined by Equation (5). As expected, the efficiency decreases with the increase of the convection coefficient; it is higher for materials with higher thermal conductivity and lower for lattices with more layers as the average distances from the channel walls increases. From these graphs, it is clear that the parameter mL is not able to combine the effect of external convection and thermal conductivity within the lattice into a single non-dimensional group as well as for simpler configurations. In fact, for a fixed geometry, Figure 9 shows that the relation between the lattice efficiency and this parameter is not biunivocal as suggested by heat transfer theory. This can be explained by two main reasons: the fluid temperature is far from being uniform along the lattice structure, as it significantly decreases from its inlet value to a temperature close to the one of the channel walls. Secondly, due to the lattice structure, both axial and transversal heat transfers are significant and the temperature field within the solid structure is three-dimensional. On the contrary, when the theory of fin efficiency is developed it is assumed that the fluid temperature is uniform and that the temperature gradients inside the solid are significant in one direction only.

Figure 10 shows how radiative heat transfer and the working temperature levels affect the lattice efficiency. Radiative heat transfer makes the lattice temperature distribution more uniform by decreasing its value at the centre of the channel, making it closer to the one of the channel walls. This in turn increases the heat exchanged by convection with the fluid. Hence, its impact is significant when the reference fin efficiency is low because of relatively low thermal conductivity (like ZrO_2_) or because of high convective heat transfer values. For example, for the 2 layers configuration with k = 2 W/(m∙K), Re = 85.45 and Pr = 0.7, the global efficiency without radiation is about 0.39 and radiative heat transfer is able to increase this quantity up to 0.72 at high temperature (T_inlet_ = 1200 °C), which implies an increase of about 85% of the heat exchanged. On the contrary, for the setup k = 46 W/(m∙K), Re = 170.9 and Pr = 0.7, for which the global efficiency without radiation is already high, about 0.81, its gain at the maximum investigated temperature is just about 0.045. This trend is similar for the 2 layers and 3 layers lattice, but this last configuration benefits more from radiation as it is characterised by more pronounced temperature distribution inhomogeneity.

## 4. Conclusions

In this work, material and architecture selections were performed to develop an innovative compact gas-to-gas heat exchanger. The lattice architecture was designed using a Kelvin cell, which provided the best compromise among effective thermal conductivity, specific surface area, dispersion coefficient and pressure loss, compared to other cell geometries. Si-SiC was selected as promising materials for the application and the ceramic lattices were then fabricated by combining polymer 3D printing with the replica method. A detailed analysis of the heat transfer characteristic of Kelvin cells-based lattice structures was performed by means of a 3D CFD-based approach. Moreover, a detailed methodology to investigate this kind of problems is described. The results of this investigation are a collection of correlations which can be used to estimate the convective heat transfer coefficient and the pressure drops of this kind of structures when they are exploited in compact heat exchangers. These correlations are valid in the ranges: 5 < Re < 220, 0.7 < Pr < 0.9 which are representative of gas-to-gas heat exchange at high temperature. Numerical results shows that radiative heat transfer plays an important role when the fin efficiency is low, while for high conductivity material like Si-SiC, the influence of this effect can be neglected.

## Figures and Tables

**Figure 1 materials-14-03225-f001:**
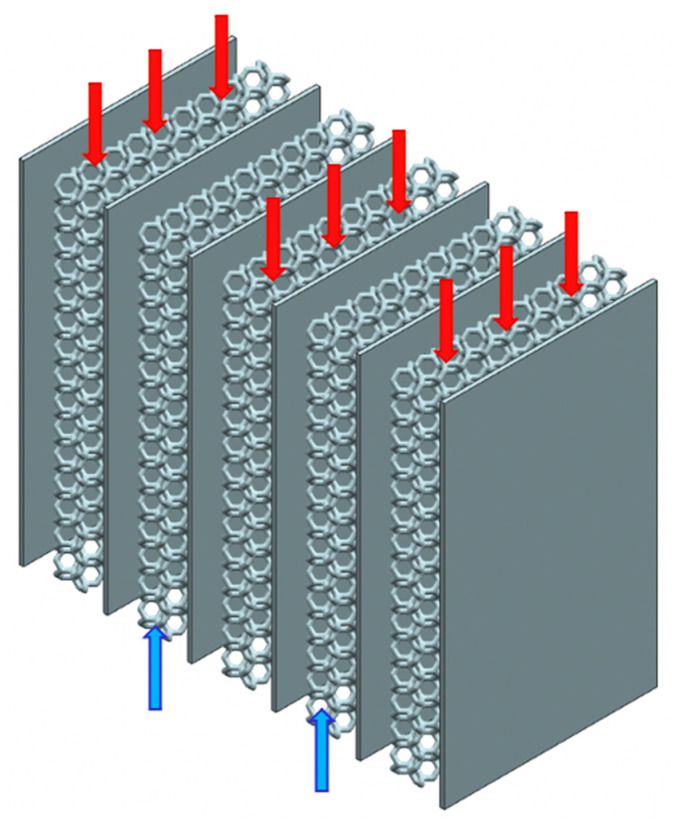
Schematic of the proposed compact gas-to-gas heat exchanger with engineered ceramic cellular architectures between the consecutive border plates.

**Figure 2 materials-14-03225-f002:**
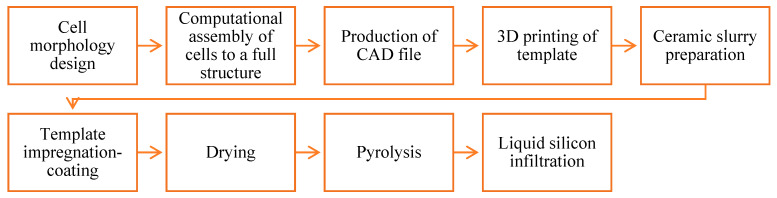
Production steps to fabricate SiSiC lattices using the replica method.

**Figure 3 materials-14-03225-f003:**
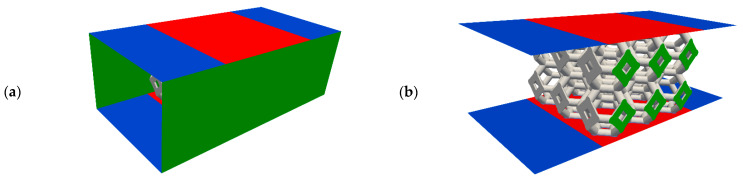
Example of computational domain: picture (**a**) shows the box surrounding the lattice. Picture (**b**) shows the same box without the lateral boundaries. The different boundary conditions are represented with different colours: blue patches are adiabatic no-slip walls. Green patches are cyclic and red patched are fixed temperature no-slip walls. Depending on the setup, the lattice surface (in grey) can be a fixed temperature wall or the interface between the fluid region and the solid one. The inlet and outlet patches are missing in both images for clarity.

**Figure 4 materials-14-03225-f004:**
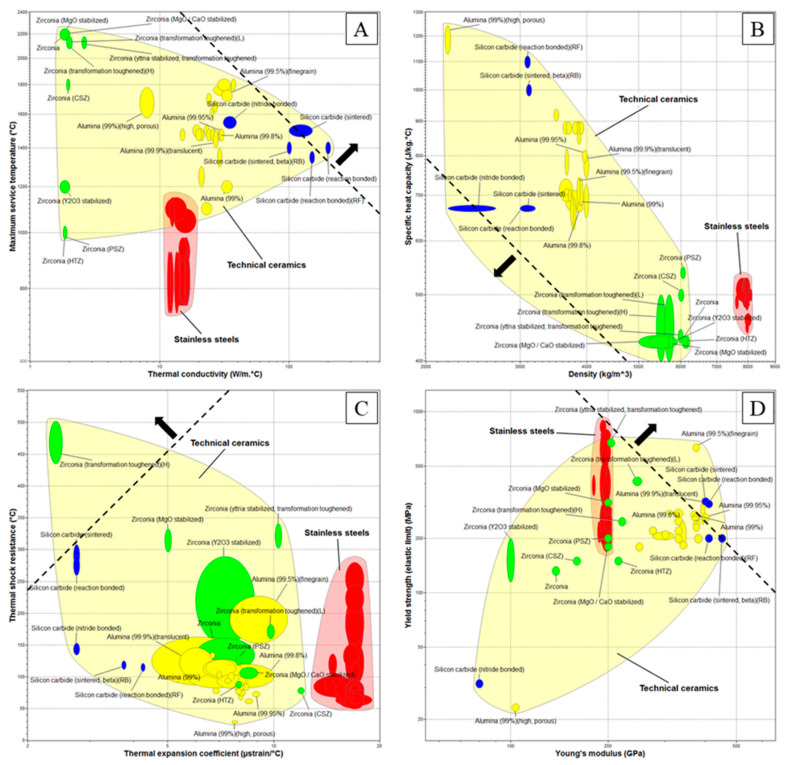
Ashby materials property chart for technical ceramics and stainless steels at 25 °C: (**A**) thermal conductivity against maximum service temperature; (**B**) density against specific heat capacity; (**C**) thermal expansion coefficient against thermal shock resistance; (**D**) Young’s modulus against yield strength. Areas’ legend: Silicon Carbide—blue; Zirconia—green; Alumina—yellow; Stainless steel—red. Materials properties were obtained from Ansys GRANTA EduPack software, ANSYS, Inc., Cambridge, UK, 2021 (www.ansys.com/materials, accessed on 30 April 2021).

**Figure 5 materials-14-03225-f005:**
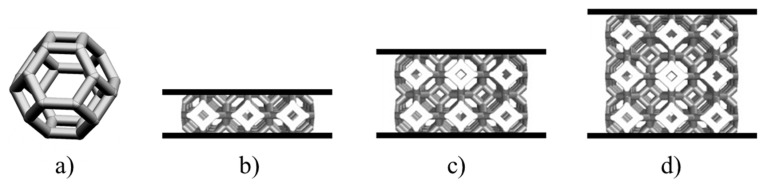
Geometry of the lattice structure investigated in this work: (**a**) Kelvin unit-cell; (**b**) 1-layer lattice; (**c**) 2-layers lattice; (**d**) 3-layers lattice.

**Figure 6 materials-14-03225-f006:**
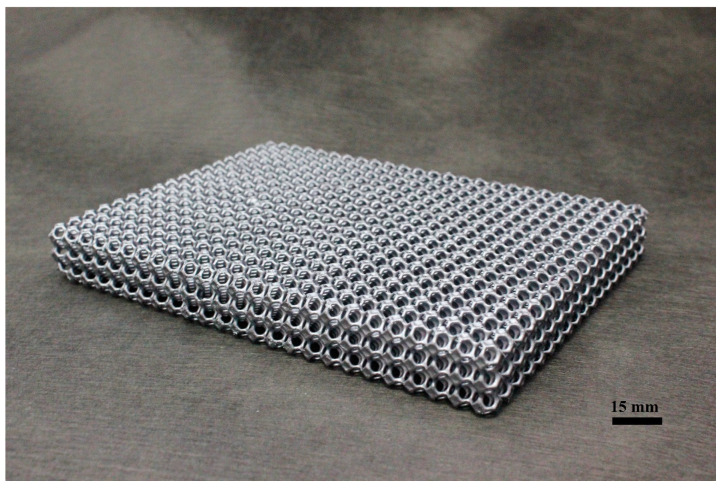
SiSiC lattice fabricated with replica method.

**Figure 7 materials-14-03225-f007:**
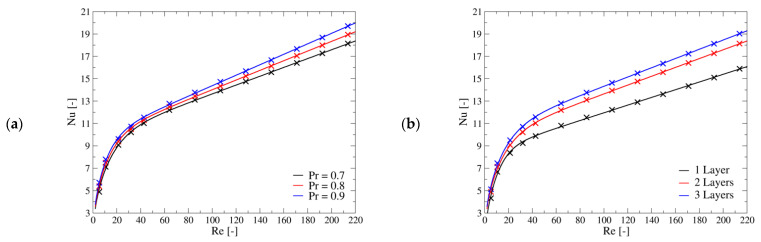
Nusselt as function of the Reynolds and Prandtl numbers. (**a**) Results related to the geometry with 2-layers of cells. (**b**) comparison of the results of the three geometries with Pr = 0.7. The cross marks are the results of CFD computation, while the continuous lines are the regression curve given by Equation (15).

**Figure 8 materials-14-03225-f008:**
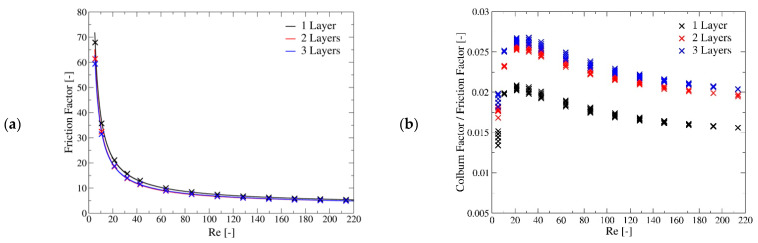
(**a**) Friction factor dependence of the Reynolds number for the investigate geometries. (**b**) Ratio of the Colburn Factor on the friction factor as function of the Reynolds number for the investigate geometries. The cross marks are the results of CFD computation, while the continuous lines are the regression curve given by Equation (17).

**Figure 9 materials-14-03225-f009:**
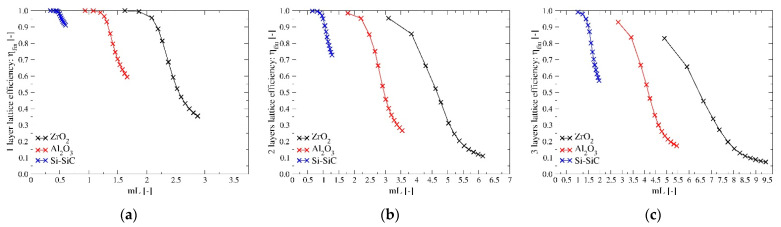
(**a**–**c**) Lattice efficiency of the three geometries for the three materials investigated in this work as a function of the non-dimensional parameter “mL” defined in Equation (5).

**Figure 10 materials-14-03225-f010:**
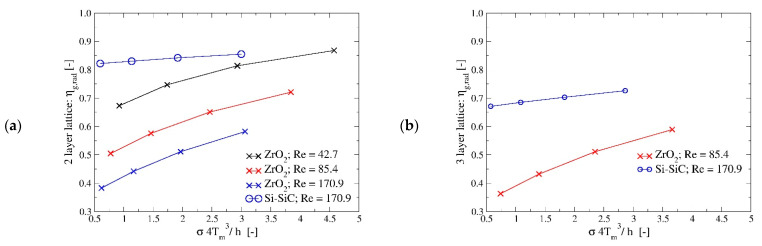
(**a**,**b**) Effect of radiative heat transfer on the fin global efficiency for the 2 and 3 layers lattice configurations. The picture compares the global efficiency with the non-dimensional group defined in Equation (6).

**Table 1 materials-14-03225-t001:** Heat exchangers operating conditions of the HYDROSOL-beyond project.

HX	Side	Parameter	Design Specification
Inert gas circuit heat exchanger	Cold side	Composition of the fluid to be heated [-]	N_2_
		Inlet temperature [°C]	600
		Mass flow rate [kg/h]	100
		Outlet target temperature [°C]	1150
		Working pressure [bar]	<2
	Hot side	Composition of the fluid [-]	N_2_ + O_2_ wt% N_2_ ≈ 99.9 wt% O_2_ ≈ 0.1
		Inlet temperature [°C]	1200
		Mass flow rate [kg/h]	100
		Outlet target temperature [°C]	Result of the design
		Working pressure [bar]	<2
Steam generation circuit heat exchanger	Cold side	Composition of the fluid to be heated [-]	H_2_O
		Inlet temperature [°C]	300
		Mass flow rate [kg/h]	10
		Outlet target temperature [°C]	800
		Working pressure [bar]	<2
	Hot side	Composition of the fluid [-]	H_2_O + H_2_ + N_2_ wt% H_2_O ≈ 94.6 wt% H_2_ ≈ 0.6 wt% N_2_ ≈ 4.8
		Inlet temperature [°C]	900
		Mass flow rate [kg/h]	10.35
		Outlet target temperature [°C]	Result of the design
		Working pressure [bar]	<2

**Table 2 materials-14-03225-t002:** Pros and cons of the replica method.

Pros	Cons
- Flexible in the choice of geometry. - Flexible in the choice of slurry and powders. - Allows to make large pieces. - Cost-efficient. - Good material properties (ceramic).	- Minimum cell size dictated by polymeric 3D printing and slurry rheology. - Possible local coating inhomogeneity.

**Table 3 materials-14-03225-t003:** Geometrical parameter of the lattice structures. The total surface is the sum of the lattice and the flat channel surfaces connected to the lattice. The specific surface is computed as the ratio of the total surface to the volume of the box (fluid + solid) in which the lattice is embedded.

Lattice Structure	Total Surface [m^2^]	Lattice Surface /Total Surface [−]	Total Surface/Surface with No Lattice [−]	Specific Surface [1/m]	Porosity [−]
1-layer (1 × 3 × 3)	1.8335 × 10^−3^	0.706	2.53	759.7	0.847
2-layers (2 × 3 × 3)	3.0284 × 10^−3^	0.760	4.17	641.9	0.844
3-layers (3 × 3 × 3)	4.3266 × 10^−3^	0.832	5.96	615.6	0.839

**Table 4 materials-14-03225-t004:** Summary of the selected boundary conditions. The colour of the patches refers to Figure 3. Note that the fixed values are not the same for all simulations, thus they are not made explicit in this table.

Field	Inlet	Outlet	Lattice Surface (Grey Colour)	Central Box Top and Bottom Patches (Red Colour)	Development Region Top and Bottom Patches (Blue Colour)
Velocity	Parabolic profile	Zero gradient	No-slip	No-slip	No-slip
Temperature	Fixed value	Zero gradient	Fixed value/interface fluid solid	Zero gradient	Fixed value
Pressure	Fixed value	Zero gradient	Fixed flux pressure	Fixed flux pressure	Fixed flux pressure

**Table 5 materials-14-03225-t005:** Coefficients of the regression curve found in Equation (15).

Lattice Structure	α	β	γ	δ	λ	Maximum Relative Error [%]	Average Relative Error [%]
1 layer (1 × 3 × 3)	8.510	0.0804	0.5344	0.0438	0.682	6.37	0.92
2 layers (2 × 3 × 3)	9.708	0.0503	0.4165	0.0502	0.679	5.66	0.62
3 layers (3 × 3 × 3)	10.197	0.0528	0.4108	0.0526	0.677	5.90	0.62

**Table 6 materials-14-03225-t006:** Coefficients of the regression curves of Equations (16) and (17).

Lattice Structure	ω	θ	ε	σ	φ	Maximum Relative Error Equation (16) [%]	Maximum Relative Error Equation (17) [%]
1 layer (1 × 3 × 3)	325.3	4.672	301.2	10.457	−1.210	14.10	3.90
2 layers (2 × 3 × 3)	329.8	3.891	319.7	6.316	−0.507		
3 layers (3 × 3 × 3)	318.5	3.872	308.6	6.253	−0.498		

## Data Availability

The data presented in this study are available in https://repository.supsi.ch/.

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
