# Peer review of "Application of Ceramic Lattice Structures to Design Compact, High Temperature Heat Exchangers: Material and Architecture Selection"

_materials, 2021, doi:10.3390/ma14123225_

Round 1
Reviewer 1 Report
This is a very nice and valuable paper which intends to develop an innovative compact gas-to-gas heat exchanger with an architecture based on Kelvin cells and Si-SiC as optimal material. However, the paper needs some minor corrections related to the presentation, as follows:
Line 149: The symbols for Reynolds and Prandtl numbers, Re and Pr, respecyvely, must be introduced here because they are used later but never defined (e.g., Eq.15)
Line 151: Eq. 2, kf needs to be defined
Line 152: the expression “…which is consistent with the porous media literature” is rather confuse, needs to be reformulated
Line 179: Eq. k and dstrut in Eq. 5 must be defined in the text
Sub-subsections 2.5.2 , 253, (Lines 210-220, 247, 258): Why quotation marks are used to define the quantities used in the equations? In the previous subsections they were not used. This looks as the result of two different authors who did not match their works. In addition, the quantities must be italicized as they appear in the corresponding equations (Eq. 10 and Eq. 11). The same occurs in the next subsections (E.g., lines 436, 439, 449)
Subsection 3.3 contains a short sentence and a single picture. Is it necessary or can be included into a more consistent subsection?
Eq. 15, and Eq.16, Eq.17 use the same regression coefficients α, β, γ, δ, λ. However, it seems that they are not identical according to Table 5 and Table 6.
References Why the short form of journals is not used? It would save space.
Reviewer 2 Report
A well argued and executed work but its main premise is weak.
There is a weakness in your use of the generic materials selection process (Ashby maps) for deciding on materials which are not represented in those maps! The maps are only for general guidance and that is why they offer properties for high density, monolithic generic SiC, Al2O3 etc, NOT SiSiC which is a wholly different material. If you really wanted to do rogorous materials selection you should have made specimens and measured the pertinent properties of SiSiC, incorporated them in the maps and then do the analysis! I bet the results would have been quite different: e.g. SiSiC has a much lower use temperature and Young's modulus than SiC but a higher thermal conductivity and toughness, etc.
Be that as it may, the overall approach and analysis of your work is good and quite rigorous.
Perhaps you should add a comment that you did the materials comparison using the Ashby Maps as a general guidance toward potentially using SiC (if you could) for HXs. But because of the difficulty in actually sintering pure SiC coatings made by repeated slurry dipping, you decided on infiltrating with Si... this does not remove the weakness of your central premise though...
Round 2
Reviewer 2 Report
Your replies on my comments are accepted, but the results on SiSiC still require careful explanation
